

# Unsupervised classification of snowflake images using a generative adversarial network and $K$-medoids classification

Jussi Leinonen[1] and Alexis Berne[1]

[1]Environmental Remote Sensing Laboratory, École polytechnique fédérale de Lausanne, Lausanne, Switzerland

**Correspondence:** Jussi Leinonen (jussi.leinonen@epfl.ch)

**Abstract.** The increasing availability of sensors imaging cloud and precipitation particles, like the Multi-Angle Snowflake Camera (MASC), has resulted in datasets comprising millions of images of falling snowflakes. Automated classification is required for effective analysis of such large datasets. While supervised classification methods have been developed for this purpose in the recent years, their ability to generalize is limited by the representativeness of their labeled training datasets,

which are affected by the subjective judgment of the expert and require significant manual effort to derive. An alternative is unsupervised classification, which seeks to divide a dataset into distinct classes without expert-provided labels. In this study, we introduce an unsupervised classification scheme based on a generative adversarial network (GAN) that learns to extract the key features from the images. Each image is then associated with a distribution of points in the feature space, and these distributions are used as the basis of $K$-medoids classification and hierarchical clustering. We find that the classification scheme

is able to separate the dataset into distinct classes, each characterized by a particular size, shape and texture of the snowflake image, providing signatures of the microphysical properties of the snowflakes. This finding is supported by a comparison of the results to an existing supervised scheme. Although training the GAN is computationally intensive, the classification process proceeds directly from images to classes with minimal human intervention and therefore can be repeated for other MASC datasets with minor manual effort. As the algorithm is not specific to snowflakes, we also expect this approach to be relevant

to other applications.

## 1 Introduction

The microphysical properties of atmospheric ice and snow have significant implications for several topics in atmospheric science. In numerical weather prediction (NWP) and climate models, the representation of ice processes has a considerable

influence on the forecast (Molthan and Colle, 2012; Morrison et al., 2015; Gultepe et al., 2017; Elsaesser et al., 2017), affecting the distribution of predicted precipitation, latent heat and radiative effects. More generally, precipitation and clouds are recognized as being among the largest uncertainties in climate predictions (e.g. Flato et al., 2013). In another context, microphysical assumptions also play an important role in the remote sensing of ice and snow because the remotely obtained signal only





conveys partial information about the properties of the icy hydrometeors, and retrieval algorithms need to be constrained by
prior knowledge about the microphysics (e.g. Delanoë and Hogan, 2010; Wood et al., 2014; Mace and Benson, 2017; Leinonen
et al., 2018).

Given the importance of microphysics, the observational geoscience community has made considerable efforts to develop
instruments that characterize the microphysical properties of falling snowflakes in situ. Measuring the properties of individual
falling snowflakes is fairly challenging, as some important properties such as the snowflake mass are not readily observable
using visual techniques due to the variation of the internal structure of snowflakes. Moreover, the widespread optical disdrom-
eters such as the Parsivel (Löffler-Mang and Joss, 2000) and the 2D Video Disdrometer (2DVD; Schönhuber et al., 2007) can
only discern a silhouette of the falling particle, unable to provide information about the surface texture. To address this issue,
snowflake imaging instruments have been actively developed in the recent years. Among these is the Multi-Angle Snowflake
Camera (MASC; Garrett et al., 2012), which employs three cameras positioned at different angles to captures images of
snowflakes illuminated by a flash as they fall through its measurement volume.

The datasets collected so far by various groups (e.g. Gaustad et al., 2015; Notaroš et al., 2016; Praz et al., 2017; Genthon
et al., 2018) show that the detailed images obtained by the MASC provide a signature of the processes that led to the formation
of each snowflake. The MASC can discern processes such as various modes of deposition growth like columns, plates and
dendrite, as well as aggregation, riming and melting (for an overview of these, see, e.g., Lamb and Verlinde, 2011). As these
processes depend on the environmental conditions in which the snowflake grew, the MASC can provide information about the
relative occurrence of these conditions in a specific snowfall event and, over longer timescales, the local climate.

While trained human observers can determine the presence of various snow growth processes from MASC images, the large
datasets collected by the MASC require computer analysis in order to derive statistically meaningful quantities of data. The
computer processing of the image data is not straightforward because much of the information is provided by shape and the
surface texture of the snowflake. Feind (2006) and Lindqvist et al. (2012), among others, previously developed algorithms
to classify ice crystals based on images from airborne probes. To enable large-scale analysis of microphysics from MASC
data, Praz et al. (2017, hereafter P17) introduced a machine-learning based classification algorithm that uses features extracted
from the images with image-processing software, providing information about the size, shape and surface patterns of each
snowflake. This algorithm can classify the snowflakes and also estimate the degree of riming and the state of melting, enabling
microphysical information to be extracted at long timescales.

The development of convolutional neural networks (CNNs) has recently greatly improved the image-recognition skill of
computers. CNNs have proved to be able to classify images based on the image data alone, without manual feature extraction.
Instead, the CNN adaptively learns the important features of images that it is trained with. Such advances could be reasonably
expected to lend themselves well to MASC data analysis, and indeed Hicks and Notaroš (2019) recently described a CNN-
based classification scheme for MASC images that achieved a performance similar to the P17 algorithm.

Supervised learning methods, such as those adopted by the above-mentioned studies, depend on the availability of an expert-
derived training dataset to train them. Obtaining such datasets is labor intensive, especially for CNNs that benefit from large
amounts of training data. Moreover, developing the training datasets is somewhat subjective as it depends on the judgement



of the expert to determine the "correct" classification for each image. The alternative is *unsupervised* classification, which

tries to organize the training data without human intervention. Unsupervised classification methods are able to operate on entire datasets without training labels, and can be less subjective than supervised classification, but are also more complex and difficult to implement as the role of the computer-based learning system in the process is much larger. Furthermore, unsupervised classification of images requires the extraction of those features of the images that are essential to classification, and ideally those features should themselves be determined in an unsupervised manner.

Unsupervised learning from image data has recently benefited from the introduction of generative adversarial networks (GANs; Goodfellow et al., 2014). GANs consist of two neural networks (usually deep CNNs), the *discriminator* and the *generator*. These are trained adversarially: The discriminator is trained to distinguish samples that belong to the training dataset from those that do not, while the generator is trained to produce samples that the discriminator considers to be part of the training data. Consequently, the generator learns to create artificial samples that strongly resemble those found in the

dataset. GANs have been recently demonstrated to be able to create realistic examples of, for example, human faces (Karras et al., 2019) and landscapes (Park et al., 2019), and have been demonstrated to be applicable to atmospheric science data analysis (Leinonen et al., 2019).

    The GAN generator is a deterministic neural network, but it can produce different outputs because it is fed random noise as an input. The random noise is sampled from a simple probability distribution such as the multivariate standard normal

distribution. Thus, the generator learns to map the simple probability distribution to the highly complex, spatially structured distribution of the image dataset. It is fairly straightforward to add another output to the discriminator, which is trained to recover all or part of the noise input of the generator, as demonstrated by the GAN variant called the *Information-Maximizing GAN* (InfoGAN; Chen et al., 2016). Accordingly, the generator can be understood as an encoder from latent variables to image samples, and the discriminator as a decoder with the approximately inverse map. The training process encourages both the

generator and the discriminator to map the latent variables to highly recognizable features of the images, thus capturing their essential properties. In the original InfoGAN paper, it was shown that InfoGAN can, in an unsupervised manner, recognize important modes of variation in its input images. Similar results have been achieved with a related GAN variant called the bidirectional GAN (Donahue et al., 2016).

    In this article, we describe a GAN trained on a dataset of MASC images and the use of the GAN-extracted latent variables

for unsupervised classification with the $K$-medoids algorithm. We achieve more consistent classification by associating each image with a distribution of points rather than a single point in the latent space. We show that this combination of a GAN and a more traditional unsupervised machine-learning algorithm can be used to classify snowflake datasets without human intervention.

    The article is organized as follows: Section 2 describes the snowflake datasets and data processing. Section 3 gives an

overview of the machine-learning methodology, and Sect. 4 describes the implementation details used for this work. Section 5 discusses the classification results for the snowflake data and presents a quantitative and qualitative evaluation of them. Finally, Sect. 6 summarizes the study.



## 2   Data

The main source of our snowflake image dataset is the deployment of a MASC in Davos, Switzerland, during 2015 and
2016. The MASC was deployed at $2450$ m above mean sea level, with a long snowy season, and was enclosed by a Double
Fence Intercomparison Reference (DFIR) setup. The dataset includes a wide variety of snowflakes, including single crystals
of different morphologies, aggregates, rimed snowflakes and graupel, as well as partially melted snowflakes. Additionally,
in order to increase the diversity of the dataset we use data from the Antarctic Precipitation, Remote Sensing from Surface
and Space (APRES3) campaign (Genthon et al., 2018), where a MASC was deployed in Dumont D'Urville on the coast of
East Antarctica. We concentrate on this joint dataset in this paper, as we did not find the results to be very different from a
classification trained only on the Davos dataset.

We first filtered out blowing snow particles from the data from APRES3, where the MASC was not shielded by a wind fence,
using the results of Schaer et al. (2019). The raw data were then processed using the processing chain described by P17. The
resulting dataset included approximately 2.1 million grayscale snowflake images, 1.9 million from Davos and 0.2 million from
APRES3. From these, we selected high-quality images as follows. We selected snowflakes whose outlines were between $16$
and $256$ pixels in diameter; with the MASC resolution of approximately $35$ μm per pixel, this corresponds to a physical size
roughly between $0.5$ and $9$ mm. The lower limit was chosen to filter out occasional artifacts and snowflakes too small to be
recognized; the rarely applied upper limit was used to ensure that the entire snowflake fits in the image. We also removed all
particles classified as "small particles" by the P17 algorithm, although not many of these remained after imposing the minimum
size. To filter out blurry snowflakes, we required that the quality index $\xi$, defined in Appendix B of P17, be at least $10$. The
minimum $\xi$ is higher than in P17 because we wanted to avoid training the GAN to generate blurry images. Furthermore, to
remove snowflakes that were not bright enough, as well as some artifacts that passed the $\xi$ test, we required that the brightest
pixel of the image be at least $0.15$ on a scale of $0$ to $1$. The final dataset used for training comprises $195006$ snowflake images,
$166981$ from Davos and $28085$ from APRES3.

Each image was downsampled by a factor of two and then centered into a $128 \times 128$ grayscale image to yield constant-sized
images, as needed for training. Many images included fairly dim areas; to make these more visible, we applied the following
transformation to the brightness:

$$
f(x) = \begin{cases} \frac{b}{a}x, & x < a \\ b + \frac{1-b}{1-a}(x-a), & x \geq a \end{cases} \tag{1}
$$

with $a = 0.1$ and $b = 0.2$ chosen, somewhat subjectively, to improve the visibility of dark snowflakes without losing too much
contrast. This maps brightness values $0 \rightarrow 0$, $a \rightarrow b$, $1 \rightarrow 1$ and linearly between these points. The resulting image dataset has
a mean brightness of $0.28$ for non-empty pixels (i.e. those of brightness $> 0$).

Following standard practice in training convolutional neural networks, we increase the diversity of training samples using
data augmentation. Before using images for training, we perform the following random augmentations to each image:

- – Rotation by an angle between $0°$ and $360°$





– Mirroring around the vertical and/or horizontal axes

– Zooming the image by a factor between 0.9 and 1.0

– Adjusting the brightness of the image by $-10\%$ to $+10\%$, truncating the brightness of each pixel afterwards between 0 and 1

– Translation from the original position by $-4$ to $+4$ pixels in the horizontal and vertical directions (the maximum shift is
fairly small to avoid pushing the image out of bounds).

The training data are available for replication purposes. The details can be found under "code and data availability" at the end of the article.

## 3 Methods

In this section, we provide a brief overview of the existing techniques we applied in our classification scheme. The specific
implementation details and novel methodology used in this study can be found in Sect. 4.

### 3.1 Convolutional neural networks

The development of the theory and best practices for CNNs has rapidly enhanced the capacity of computers to process spatially structured data such as images (LeCun et al., 2015). These networks employ a series of convolution operations and nonlinearities to extract successively higher-level features of images. Each such operation is called a *layer* of the network. The most
common types of layers are described briefly below:

**Dense layers** (also called fully connected layers) map their input vector $\mathbf{x}$ to the output $\mathbf{y}$ as an affine transformation $\mathbf{y} = \mathbf{W}\mathbf{x} + \mathbf{b}$ where the matrix $\mathbf{W}$ and vector $\mathbf{b}$ consist of trainable parameters.

**Convolution layers** map the channels of their input to their output as a sum of convolution operations. The convolution kernels are trainable parameters that are learned by the network.

**Activation layers** apply a nonlinear function to their input. This, in combination with the mixing operation implemented by the convolution and/or dense layers, allows the network to learn highly nonlinear maps when enough layers are used. The most common activation function in CNNs is the rectified linear unit (ReLU; Nair and Hinton, 2010), which is defined as

$$a(x) = \begin{cases} 0, x < 0 \\ x, x \geq 0 \end{cases} \tag{2}$$





In this work, we use a variant called Leaky ReLU:

$$a(x) = \begin{cases} \alpha x, x < 0 \\ x, x \geq 0 \end{cases} \qquad (3)$$

where the hyperparameter $\alpha$ is a small positive number that is used to permit a small but nonzero gradient at $x < 0$.

**Pooling layers** reduce the spatial dimensions of their input by applying a pooling operation such that each $M \times N$ neighborhood of the input is mapped to a single value. Usually, either the average or the maximum of the inputs is used as the
pooled value. Pooling operations can sometimes be replaced by strided convolutions, which skip some points (e.g. every other point) of the input to reduce the spatial dimensionality of the output.

**Normalization layers** seek to normalize their input data, usually trying to constrain the input to optimal mean and variance (which are typically either fixed to $0$ and $1$, respectively, or optimized as parameters of the network). This seeks to keep the variables in the active (i.e. nonzero gradient) ranges of the activation functions and to reduce the dependence be-
tween the parameters of the network. Common normalization strategies include batch normalization (Ioffe and Szegedy, 2015) and instance normalization (Ulyanov et al., 2016). More recently, Huang and Belongie (2017) introduced adaptive instance normalization (AdaIN), which allows the network to be adapted to different styles through external weighting of layers.

In practice, layers are often organized in *blocks*, predefined series of operations implemented using the above-mentioned
types of layers. Recently, residual network (or "ResNet") blocks, which add their input to their output allowing the nonlinear part to operate only on the residual, have been popular after it was found that they improve accuracy over similar-sized non-residual networks in classification tasks (He et al., 2016).

Neural networks are trained by minimizing a loss function through gradient descent, where gradients are evaluated using the backpropagation algorithm (Rojas, 1996). For more comprehensive and technical introductions to CNNs, we refer the reader
to, for example, Goodfellow et al. (2016).

### 3.2 Generative adversarial networks

A GAN is a system of two neural networks, usually CNNs, that are trained adversarially, i.e. competing against each other. One network, the discriminator $D$, is a binary classifier that is optimized to distinguish inputs that belong to the training dataset ("real") from those that do not ("fake"). The other network, the generator $G$, is simultaneously trained to produce
artificial outputs that the discriminator considers to be real. Thus, the generator learns to produce examples that look real to the discriminator, and hence resemble the examples found in the training dataset. A sufficiently large dataset is needed such that neither the discriminator nor the generator can simply memorize the set of input images, but must instead learn the structure of the inputs. The input to the generator, $\mathbf{z}$, is sampled from a simple probability distribution; we use a multivariate standard normal distribution in this work.





The initially proposed GAN loss (Goodfellow et al., 2014) treats $D$ as a probabilistic binary classifier where the output $D(\mathbf{x}) \in [0,1]$, representing the estimated probability of $\mathbf{x}$ being a fake rather than a real sample. Using binary cross entropy, the GAN losses can then be written as

$$L_D(\mathbf{x}, \mathbf{z}; \theta_D) = \log(D(\mathbf{x})) - \log(1 - D(G(\mathbf{z}))) \tag{4}$$

$$L_G(\mathbf{z}; \theta_G) = \log(1 - D(G(\mathbf{z}))). \tag{5}$$

Where $L_D$ is the discriminator loss and $L_G$ the generator loss, and $\theta_D$ and $\theta_G$ are the corresponding trainable weights. The optimization goals are:

$$\min_{\theta_D} \quad \mathrm{E}_{\mathbf{x}, \mathbf{z}}[L_D(\mathbf{x}, \mathbf{z}; \theta_D)] \tag{6}$$

$$\min_{\theta_G} \quad \mathrm{E}_{\mathbf{z}}[L_G(\mathbf{z}; \theta_G)]. \tag{7}$$

As the goals are mutually contradictory, the training proceeds by alternating between training $D$ with fixed $\theta_G$, and training $G$
with fixed $\theta_D$.

More recently there have been developments that seek to provide a more stable loss for GANs. The Wasserstein GAN (WGAN; Arjovsky et al., 2017) is motivated by the Wasserstein distance between probability distributions, and can be written simply as

$$L_D(\mathbf{x}, \mathbf{z}; \theta_D) = D(\mathbf{x}) - D(G(\mathbf{z})) \tag{8}$$

$$L_G(\mathbf{z}; \theta_G) = D(G(\mathbf{z})). \tag{9}$$

Thus with WGAN, the discriminator is trained to make its output as small as possible for real inputs, and as large as possible for generated inputs. A slight variant called the "hinge loss" was employed to regularize the discriminator by Donahue and Simonyan (2019) for very high image quality in the context of unsupervised feature learning:

$$L_D(\mathbf{x}, \mathbf{z}; \theta_D) = h(-D(\mathbf{x})) + h(D(G(\mathbf{z}))) \tag{10}$$

$$h(t) = \max(0, 1 - t). \tag{11}$$

WGANs can be constrained by combining them with gradient penalty, which regulates the gradients of the discriminator outputs, with respect to the inputs, by penalizing them for deviating from unit length (Gulrajani et al., 2017). This yields an additional term for the discriminator loss as

$$L_{\mathrm{GP}}(\mathbf{x}, \mathbf{z}; \theta_D) = (||\nabla_{\hat{\mathbf{x}}} D(\hat{\mathbf{x}})||_2 - 1)^2. \tag{12}$$

The samples $\hat{\mathbf{x}}$ are randomly weighted averages between real and generated samples:

$$\hat{\mathbf{x}} = \epsilon \mathbf{x} + (1 - \epsilon) G(\mathbf{z}) \tag{13}$$

where $\epsilon$ is a random number sampled from the uniform distribution between $0$ and $1$.

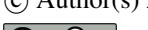



### 3.3 Latent variable extraction with GANs

The generator mapping from the simple probability distribution of $\mathbf{z}$ to the complex distribution of $\mathbf{x}$ is not invertible in the
basic GAN formulation. However, mapping $\mathbf{x}$ to $\mathbf{z}$ is of great interest to unsupervised learning as it allows the essential features
of the training images to be encoded into much simpler vectors in the latent distribution $\mathbf{z}$. Consequently, several GAN variants
have been proposed that incorporate an approximate inverse mapping from $\mathbf{x}$ to $\mathbf{z}$ using various approaches (e.g. Chen et al.,
2016; Donahue et al., 2016; Ulyanov et al., 2017).

### 3.4 Classification: $K$-means and $K$-medoids

$K$-means and $K$-medoids are unsupervised classification methods that seek to associate each point in a dataset with one of
$K$ centerpoints (Kaufman and Rousseeuw, 1990; Jain et al., 1999). Thus, data points that are close to each other tend to be
associated with the same centerpoint and thus they can be considered to be members of the same class.

The $K$-means and $K$-medoids algorithms both select the centerpoints $\mathbf{c}_j$, $j = 1..K$, to minimize the cost

$$L \quad = \quad \frac{1}{N} \sum_{i=1}^{N} d(\mathbf{z}_i, \mathbf{c}_{\mathrm{n},i}) \tag{14}$$

$$\mathbf{c}_{\mathrm{n},i} \quad = \quad \underset{\mathbf{c}_j}{\operatorname{argmin}} \; d(\mathbf{z}_i, \mathbf{c}_j) \tag{15}$$

where $d$ is a distance metric between two points. In other words, they minimize the distance of each point $\mathbf{z}_i$ to its nearest
centerpoint $\mathbf{c}_{\mathrm{n},i}$. The standard $K$-means algorithm uses the squared Euclidean distance (SED) metric

$$d_{K\text{-means}}(\mathbf{z}_1, \mathbf{z}_2) = |\mathbf{z}_1 - \mathbf{z}_2|^2 \tag{16}$$

It is called $K$-means because it can be seen as a generalization of the $K = 1$ case where the unique optimal centerpoint is simply
the mean of the data points. In contrast to that special case, with $K \geq 2$ the solution is less straightforward and must be found
iteratively. The standard algorithm is to start with randomized centerpoints and repeat the following steps until convergence:

1. Associate each data point $\mathbf{z}_i$ with a nearest centerpoint $\mathbf{c}_{\mathrm{n},i}$.

2. Update each $\mathbf{c}_j$ to the mean of data points associated with it.

This iteration is not guaranteed to find the globally optimal solution; restarting the algorithm multiple times and selecting the
best solution (smallest $L$) is often helpful. Furthermore, the number of centerpoints must be set manually; it is not inferred by
the algorithm.

The $K$-medoids algorithm can be understood a variant of $K$-means. Unlike with $K$-means, where a centerpoint can be
an arbitrary point in space, $K$-medoids select $K$ data points to act as centers ("medoids"). The advantage of this is that an
arbitrary distance metric $d$ between points can then be used. The partitioning around medoids algorithm (PAM; Kaufman and
Rousseeuw, 1990) starts with randomly selected medoids and finds the optimum by iterating the following steps:

1. For each medoid $\mathbf{c}_j$:





    (a) Compute $L_{ij}$ as the values that $L$ would have if $\mathbf{z}_i$ was used as $\mathbf{c}_j$ instead.

    (b) If the cost would improve, i.e. $\min_i L_{ij} < L$, set $\mathbf{c}_j := \mathbf{z}_{i,\text{opt}}$ where $i_{\text{opt}} = \operatorname{argmin}_i L_{ij}$.

  2. If $L$ decreased, repeat. Otherwise, terminate.

As with $K$-means, achieving the global optimum is not guaranteed and restarting may be necessary.

# 4 Implementation

## 4.1 GAN architecture and training

Following Donahue and Simonyan (2019), we implement the GAN using the hinge loss (Eq. 10) and gradient penalty (Eq. 12) for the discriminator and the WGAN loss (Eq. 7) for the generator. We also add a second output $D_E$ to the discriminator to

extract the latent variables received by the generator. To enforce the recovery of the latent variables, we add a mean-square-error (MSE) loss between the original and recovered latent variables to both $D$ and $G$, similar to InfoGAN:

$$L_{\text{MSE}}(\mathbf{x},\mathbf{z};\theta_D,\theta_G) = |D_E(G(\mathbf{z})) - \mathbf{z}_{\text{l}}|^2. \tag{17}$$

where $\mathbf{z}_{\text{l}}$ is the subset of $\mathbf{z}$ that we attempt to recover. The final losses we use are then:

$$L_D(\mathbf{x},\mathbf{z};\theta_D) = h(-D(\mathbf{x})) + h(D(G(\mathbf{z}))) + \gamma L_{\text{GP}}(\mathbf{x},\mathbf{z};\theta_D) + \beta L_{\text{MSE}}(\mathbf{x},\mathbf{z};\theta_D,\theta_G) \tag{18}$$

$$L_G(\mathbf{x},\mathbf{z};\theta_G) = D(G(\mathbf{z})) + \beta L_{\text{MSE}}(\mathbf{x},\mathbf{z};\theta_D,\theta_G) \tag{19}$$

where $\gamma$ is the weight of the gradient penalty and $\beta$ is the weight of the MSE loss. We use $\gamma = 10$ following Gulrajani et al. (2017) and $\beta = 1$ as we found that this simple choice led to good convergence of both the GAN image generation and the latent-variable recovery.

    The discriminator (Fig. 1a) is implemented as a series of six ResNet-type blocks of layers, each consisting of two activation-

convolution blocks. Those blocks transform the image to 512 feature maps of $4 \times 4$ size. The convolutional blocks are followed by a pooling layer and two fully connected layers. From these, we derive the discriminator output $D$ and the latent-encoder output $D_E$. Spectral normalization (Miyato et al., 2018) is used in all layers of the discriminator. We also use the batch-statistics technique suggested by Karras et al. (2019) to encourage variety and inhibit mode collapse, a failure where the generator always generates the same (or one of a few) outputs.

The generator (Fig. 1b) derives from the StyleGAN concept of Karras et al. (2019). The network consists of five styling blocks, each of which upscales the image and then processes it with two subblocks each containing an AdaIN layer, an activation layer and a convolution layer. Deviating from Karras et al. (2019), the styling blocks are also residual, adding the result to the input at the end of processing, as we found that this improved convergence. The generator upscales an initial feature map, affine transformed from the latent variables, into a $128 \times 128$ image. The noise $\mathbf{z}$ passed to the generator consists of two

distinct components: the style that is the same for each AdaIN layer, and the additive noise that is independent to each of those layers. We use the style as the latent variable $\mathbf{z}_{\text{l}}$ that is recovered by the discriminator, while the additive noise is not recovered.





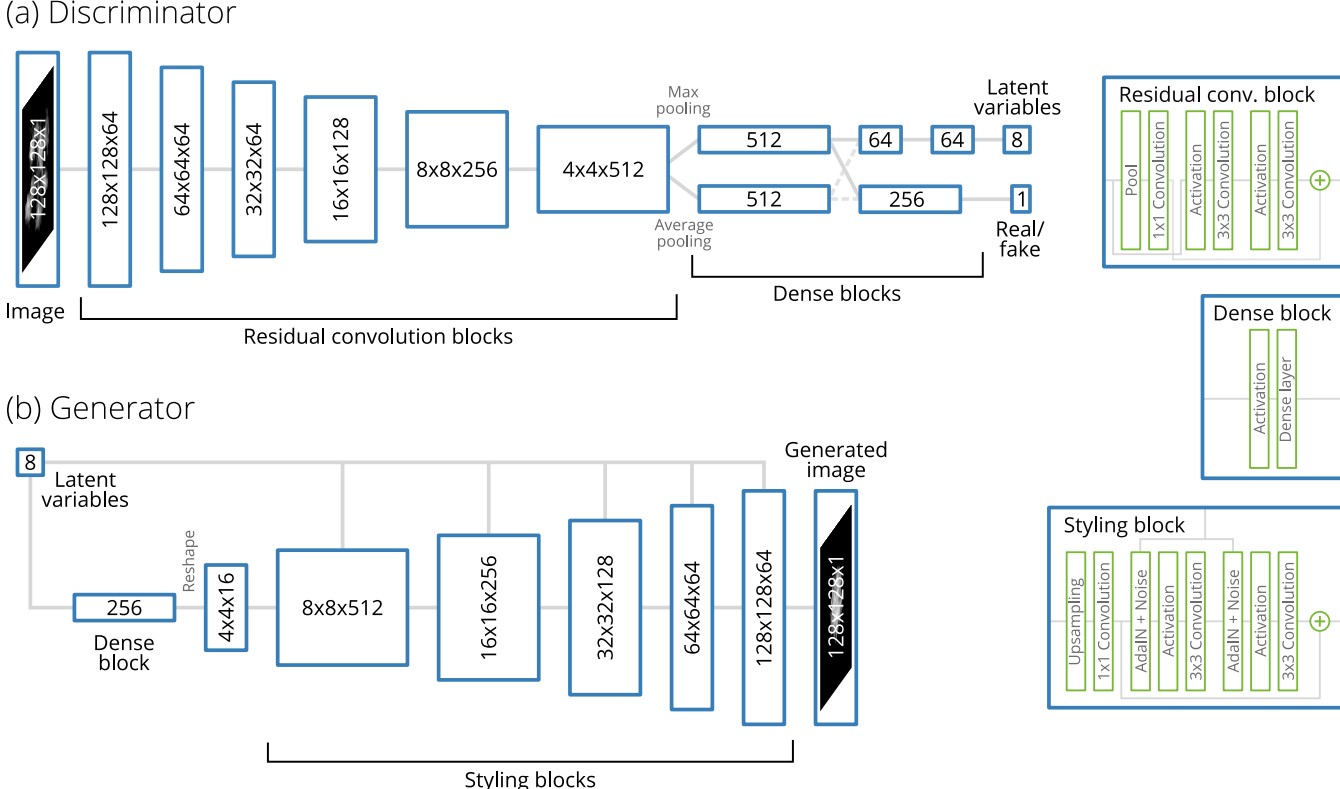

**Figure 1.** Outlines of the architectures of the (a) generator and (b) discriminator networks. The numbers on the blocks denote the size and number of feature maps; for example, $16 \times 16 \times 256$ indicates 256 feature maps of $16 \times 16$ pixels each. The number of features for each layer was determined by adapting network architectures used in earlier GAN studies, then tuning the sizes to achieve a reasonable compromise between performance and computational cost.

The objective of this is to capture the essential features of the snowflake in the style vector, while the less significant variation between individual snowflakes of the same type is represented in the noise. However, contrary to Karras et al. (2019), we found the additive noise to have an insignificant effect on the final generated images.

We train the network by alternating between training the discriminator on a single batch of data with the generator weights held constant, and training the generator with a single batch with constant discriminator weights. We use a batch size of $64$ for both networks. The training was monitored manually and terminated when neither the losses nor the image quality were any longer changing appreciably. The transformations and augmentations described in Sect. 2 were applied to each batch before processing. We used the Adam optimizer (Kingma and Ba, 2014) with learning rate set to $10^{-4}$ for both the discriminator and

the generator. We trained the network with Nvidia Tesla P100 and K40 graphics processing units (GPUs); the final training used to derive the results presented in this article took approximately $72\,\mathrm{h}$ on the K40 while training on the P100 was roughly twice as fast.



## 4.2 Clustering

In principle, we could use $K$-means classification to derive classes directly from the extracted latent variables $\mathbf{z}$. However,
we found that the latent variables code for some features that we do not actually want to use for classification: For example, some of the variation in $\mathbf{z}$ corresponds to the orientation of the particle, while a given particle should belong to the same class regardless of the orientation at which it is seen.

We make the unsupervised classification more robust and approximately invariant to unwanted features by producing random variations of each image and then associating each image with a *distribution* of points rather than a single point in the latent
space. Specifically, we perform the random augmentations described in Sect. 2 (none of which should affect the classification) to each image 100 times, extracting the latent variables $\mathbf{z}$ for each sample. From these samples, we compute the mean $\boldsymbol{\mu}_i$ and covariance $\boldsymbol{\Sigma}_i$ of the latent variables for each image $i$. Now, we can define an approximately augmentation-invariant distance between two snowflake images using a metric for the distance between two probability distributions. For this, we use the Bhattacharyya distance between two multivariate normal distributions (Fukunaga, 1990):

$$d_{\mathrm{B}}(\mathbf{x}_i, \mathbf{x}_j) \quad = \quad \frac{1}{8}(\boldsymbol{\mu}_i - \boldsymbol{\mu}_j)^T \boldsymbol{\Sigma}^{-1}(\boldsymbol{\mu}_i - \boldsymbol{\mu}_j) + \frac{1}{2} \ln\left(\frac{\det \boldsymbol{\Sigma}}{\sqrt{\det \boldsymbol{\Sigma}_i \det \boldsymbol{\Sigma}_j}}\right) \tag{20}$$

$$\boldsymbol{\Sigma} \quad = \quad \frac{\boldsymbol{\Sigma}_i + \boldsymbol{\Sigma}_j}{2} \tag{21}$$

While the distribution of latent-space points for each particle is not guaranteed to be normal, we adopt this distance metric because it is reasonably fast to compute and symmetric with respect to a swap of $i$ and $j$. It is also a generalization of the SED (Eq. 16) in the following sense: If $\boldsymbol{\Sigma}_i = \boldsymbol{\Sigma}_j = a\mathbf{I}$ for any $a \neq 0$, $d_{\mathrm{B}}$ is linearly proportional to the SED between the means,
$|\boldsymbol{\mu}_i - \boldsymbol{\mu}_j|^2$.

Additionally, we adopt a simple hierarchical clustering algorithm that can be used to iteratively reduce the number of classes derived from the $K$-medoids algorithm by forming a binary tree structure. This algorithm proceeds bottom-up as follows:

1. Start with $K$ separate branches, each containing one of the medoids.

2. On each iteration:

(a) Find the pair of branches $(j, k)$ with the shortest distance $d(j, k)$ between the two branches.

    (b) Record $j$ and $k$, then merge branch $k$ into branch $j$.

The iteration can be continued until all branches have been merged into a single tree structure. The distance between two branches, $d(j, k)$, can be defined in multiple ways, but we found that the best results were obtained by defining it as the average of all possible $d_{\mathrm{B}}$ between medoids belonging to branch $j$ and those belonging to branch $k$.
After classification, the medoids are reordered manually such that the tree structure resulting from the hierarchical clustering can be visualized clearly. This change is purely cosmetic and does not affect any metrics.

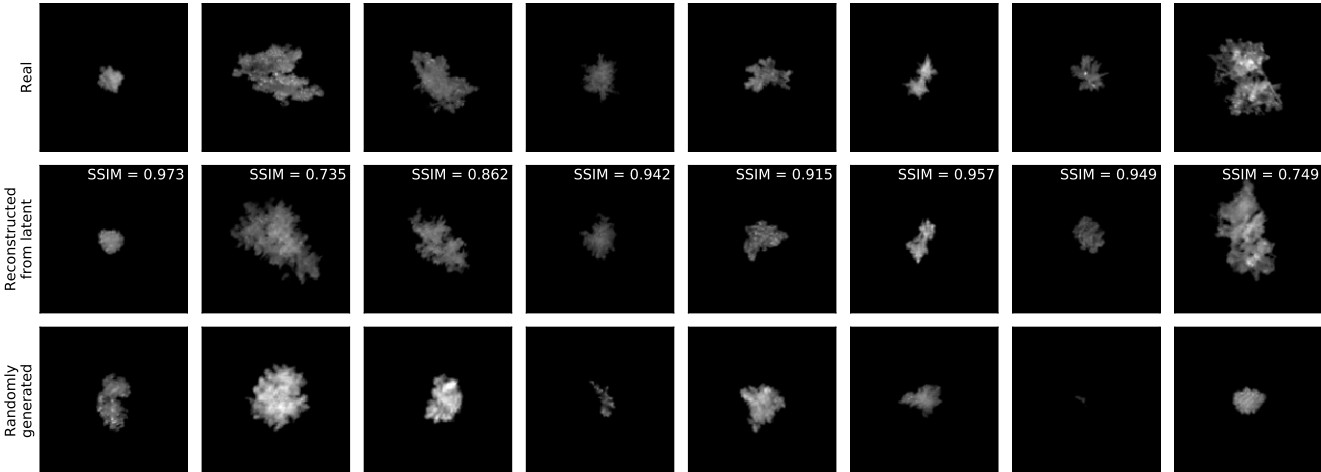

**Figure 2.** Samples of real and generated images. Top row: real images from the dataset. Middle row: images generated by extracting the latent variables from the corresponding top-row image, then generating an image with the GAN generator from that image; the SSIM between the real and generated images is also shown. Bottom row: Samples of snowflake images generated from random latent codes.

## 5 Results and discussion

### 5.1 Image generation from latent variables

Figure 2 shows samples of real and generated images. The middle row of Fig. 2 displays a reconstruction of the real snowflake shown on the top row. The reconstructed image is obtained by extracting the latent variables from the original image with $D_E$ and then generating an image from the latent variables using $G$. The reconstruction is not perfect — nor can that be expected given that $D_E$ compresses each $128 \times 128$ pixel image to only eight real numbers — but the reconstructed images are quite similar to the corresponding originals, showing that the latent codes encapsulate information about the essential features of the image such as size, shape, contrast, orientation and texture. The bottom row of Fig. 2 shows snowflakes generated using randomly selected latent variables. These images also look qualitatively plausible, demonstrating that the generator is not dependent on the latent variables being extracted from a real image.

The generator does not replicate all features found in the original snowflake. For instance, fine details of the large aggregate in the rightmost column of Fig. 2 are not reproduced in the reconstructed image. This is consistent with the relatively low structural similarity index (SSIM; Wang et al., 2004) of $0.749$ between these two images. The average structural similarity index between real and generated images in the dataset is $0.928$. However, in contrast to most applications of GANs, in this study the image generation is merely a byproduct of the classification scheme. The primary goal is to train the discriminator to extract latent variables that can be used for classification.

In Fig. 3, we show the effect of varying the latent variables on the generated image. Two latent variables are varied while the rest are held constant. This gives an example of how the GAN maps the latent variables to the data distribution. The generated





images look plausible at each combination of the two latent variables, while the image changes smoothly between two different shapes of aggregate (top left and right corners), a large rimed column-like snowflake (bottom left corner), and a small irregular snowflake (bottom right corner). This ability of the GAN to learn an encoding between the latent variables and the essential image properties is the basis of the classification.

## 5.2   Classification

### 330   5.2.1   The number of classes

A common problem with unsupervised classification is selecting the number of classes. Using the latent variables extracted by the GAN, we ran the $K$-medoids classification, as described in Sect. 4.2, for values of $K$ between 1 and 20 and recorded the change in the cost function $L$ (Eq. 15). Since the computational complexity of the $K$-medoids algorithm scales as $\mathcal{O}(N^2)$, and therefore it would be very expensive to run it for the entire dataset, we subsampled 2048 random images from the dataset

(the same subset was used for each $K$, but similar classification results were obtained with different subsets). The best solution for each $K$ was found by running the algorithm until convergence, then restarting it repeatedly until 8 restarts were performed without $L$ decreasing. The solution with the smallest $L$ was selected and the others discarded.

    The loss, as a function of $K$, is shown as the blue solid line of Fig. 4. With clustered data, such analysis often reveals the appropriate number of medoids, as $L$ decreases sharply until the number of medoids reaches the number of clusters, and much

more slowly afterwards. In our case, no such threshold is apparent. Instead, the loss decreases gradually and monotonically as $K$ increases, with diminishing returns at higher $K$.

    Given that examining $L$ as a function of $K$ does not suggest any obvious choice for the number of medoids, we need to select it subjectively. However, we can start with a relatively large $K$ and iteratively simplify the classification using hierarchical clustering, merging nearby classes. We show the results of this on the cost function on the orange dashed line (starting from

$K = 16$) and the green dotted line (starting from $K = 6$) of Fig. 4. For the purpose of calculating the cost, we selected a new medoid for each merged class as the member of the class that minimizes the sum of distances $d_{\mathrm{B}}$ from the medoid to the class members.

### 5.2.2   Characteristics of classes

Since there did not appear to be clear reasons to prefer any of the large values of $K$ over the others, we chose $K = 16$ somewhat

arbitrarily as a value that gives good intra-class similarity of the snowflakes while keeping the classes distinguishable from each other. We show samples of class members from the 16-class classification, along with the class hierarchy tree derived using hierarchical clustering, in Fig. 5. The class distance matrix, showing the Bhattacharyya distance between the class medoids, is shown in Fig. 6. Each class contains snowflakes with similar size, shape and texture.

    The hierarchical clustering groups the classes into three main branches (classes 1–5, 6–12 and 13–16, respectively). The

systematic difference between these branches is in the size of the snowflakes, while the variability within each branch reveals differences in the structure of the snowflakes.

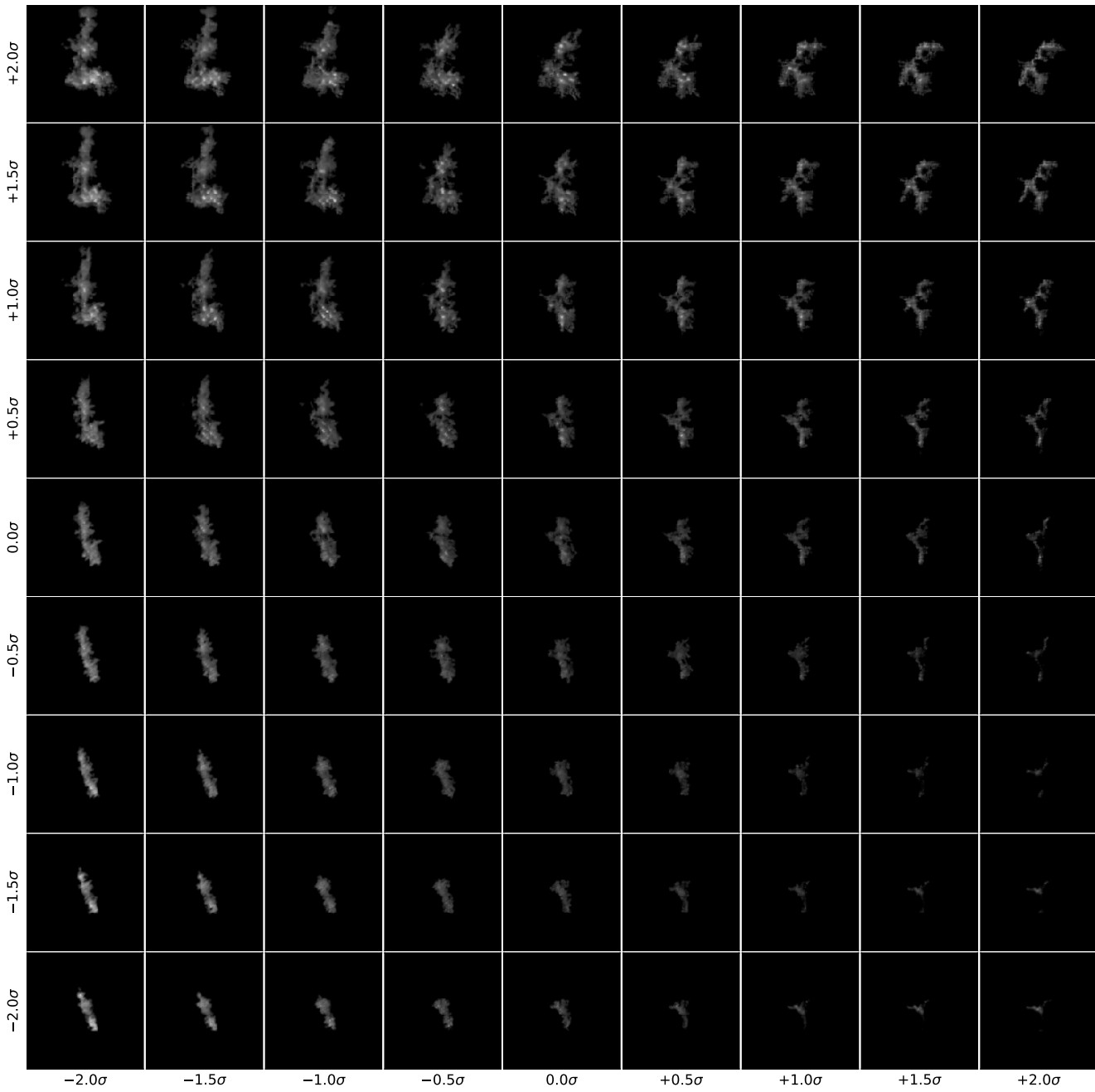

**Figure 3.** An example of the effect of varying the latent variables. Two latent variables of the GAN are varied from $-2$ to $+2$ standard deviations ($\sigma$) while the others are held constant.



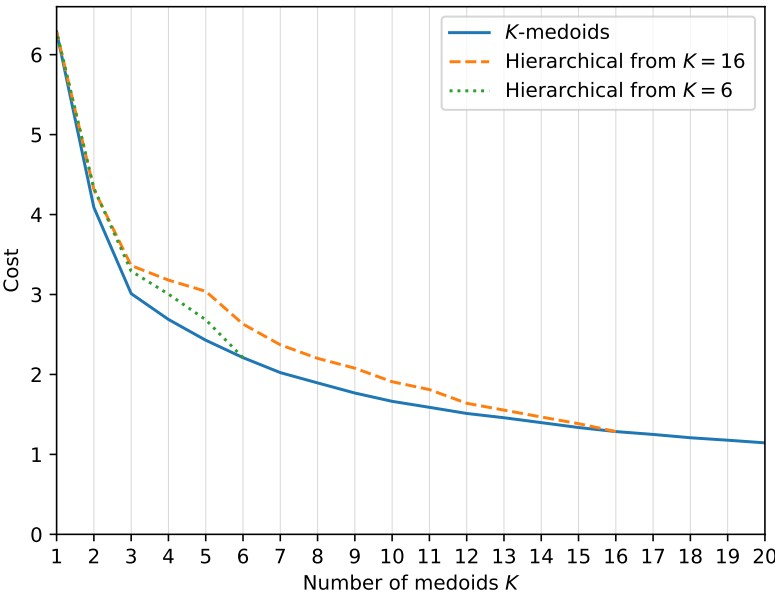

**Figure 4.** The behavior of the $K$-medoids loss $L$ as a function of the number of medoids $K$. Blue solid line: the results of the $K$-medoids algorithm for each $K$. Orange dashed line: the result of starting with the $K = 16$ result and applying hierarchical clustering (as described in Sect. 4.2). Green dotted line: as above, but starting with $K = 6$.

The first branch (classes 1–5) consists mostly of large and medium-sized aggregates with some large single crystals. Classes 1, 2 and 5 are composed mainly of moderately rimed aggregates with the main differences among these being size and complexity, both decreasing from class 1 to 2 and further from 2 to 5. Snowflakes in class 3 are highly complex but less rimed than those of class 1, while those in class 4 are the most heavily rimed in the first branch. The distances among these classes are all relatively short, and they stand out in the top-left corner of Fig. 6.

The second branch (classes 6–12) contains various ice particles smaller than those in the first branch. Classes 6 and 7 contain mostly heavily rimed snowflakes, including graupel, while 8–10 are made up of small aggregates and irregular snowflakes, those in 8 and 9 being of similar size and those in 10 somewhat smaller. The distances among these classes are very short. Class 11 resembles the classes of the first branch (as evidenced by its short distance to classes 3 and 5), containing medium-sized aggregates with little or no riming. Finally, class 12 is similar to 11, but with slightly smaller snowflakes.

The smallest particles are found in the third branch (classes 13–16). Classes 13 and 15 contain small rimed crystals and graupel, while class 14 differs from those by being unrimed or lightly rimed. Class 16 is the most visually distinct of all classes and is composed mainly of columnar crystals.

Figure 7 shows the memberships in each class. Fewer snowflakes are classified into the more extreme classes consisting of either very large or very small snowflakes. The APRES3 data contains larger proportions of the small-snowflake classes. There

**Figure 5.** Samples from each class using the 16-class classification. Each row corresponds to a class; the first column shows the particle used as the medoid while the other columns show random samples. The lines on the left illustrate the tree structure derived with hierarchical clustering.

are also more rimed snowflakes in the APRES3 data, consistent with the observations of Del Guasta et al. (1993) and Grazioli et al. (2017) that mixed-phase clouds and riming were frequent in Dumont D'Urville.





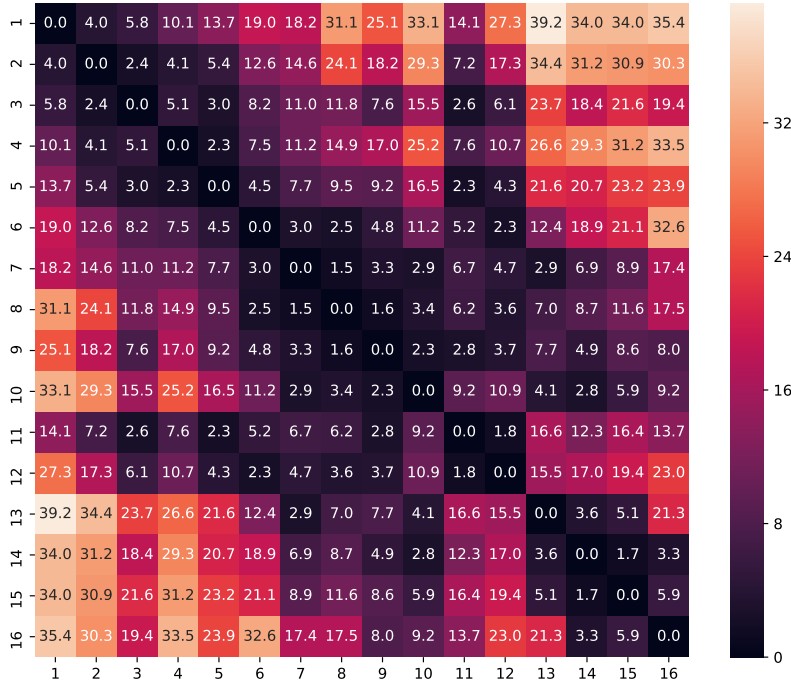

**Figure 6.** Class distance matrix for the 16-class classification, showing the Bhattacharyya distance between class medoids.

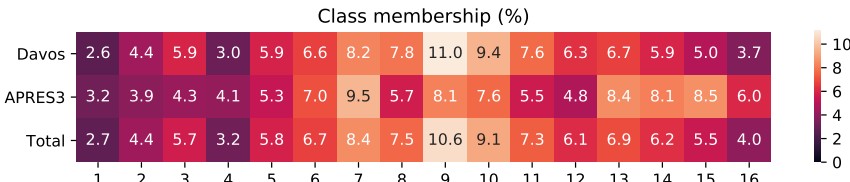

**Figure 7.** The percentage of class memberships in each class for the Davos and APRES3 data as well as the entire dataset.

In Fig. 8, we show the same type of classification as Fig. 5, but performed with only 6 classes in order to see how the
classes combine. While the columnar crystals are again well separated from other types, there is considerably more variability within each class. Particularly importantly, various degrees of riming become more mixed within the classes. Therefore, if one wants to derive information about the microphysics, especially riming, it appears to be preferable to start with a relatively large number of classes and merge them later either in an automated fashion (e.g. the hierarchical clustering shown here) or more subjectively (see Sect. 5.2.3 below). Accordingly, we will concentrate on the 16-class scheme in the rest of this article. The
other classifications can be found in the released data.





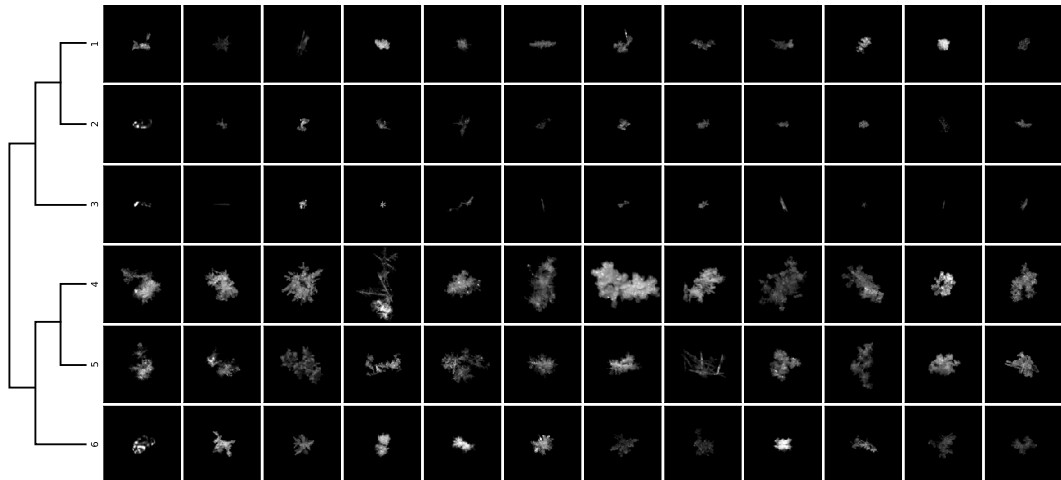

**Figure 8.** As Fig. 5, but with $K = 6$.

### 5.2.3 Microphysical classification

As mentioned above, it seems preferable to perform the classification with a fairly large number of classes and then merge them as needed. While this can be done in an objective fashion using the hierarchical clustering, the algorithms perform the analysis based only on the image properties and have no knowledge of the underlying microphysical processes. Therefore,

subjective categorization of the classes based on expert analysis can also be helpful, although it adds a supervised component to the classification process. Different applications may require different definitions, but for characterizing snowfall events we suggest the following categories:

1. *Lightly rimed aggregates*: Classes 3, 8, 9, 10 and 11

2. *Moderately rimed aggregates*: Classes 1, 2, 5 and 12

3. *Large, heavily rimed snowflakes*: Classes 4, 6, and 7

4. *Small, heavily rimed snowflakes*: Classes 13 and 15

5. *Small crystals and their aggregates*: Class 14

6. *Columnar crystals*: Class 16

### 5.2.4 Comparison to supervised classification

Figure 9 shows the corresponding P17 classes for each of our 16 classes, normalized such that the membership counts sum to 100% for each of our classes. This categorization is generally consistent with the analysis of the microphysical properties that we have presented above, showing that the two different schemes are sensitive to many of the same features in the images.





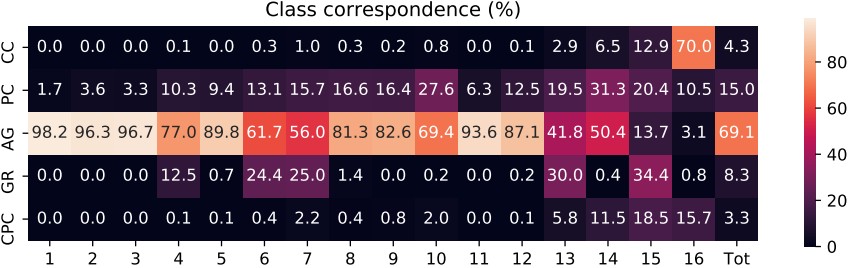

**Figure 9.** A matrix showing the correspondence between our 16-class classification and the P17 classes. Each column has been normalized to sum to 100%, and the last column shows the class memberships in the entire dataset. The P17 classes are abbreviated as follows: CC (columnar crystals); PC (planar crystals); AG (aggregates); GR (graupel), CPC (combinations of plates and columns).

The classes designated as aggregates (categories 1 and 2 above) are also dominated by aggregates in Fig. 9. The heavily rimed classes (categories 3 and 4) contain, as expected, more graupel than the other classes, although in all of these except for class 15 the aggregates are actually the most common type. This is because aggregates are overrepresented in the dataset as a whole, and also because the P17 scheme draws a distinction between heavily rimed aggregates and graupel, which may be difficult to distinguish in practice. Class 14 is a fairly generic grouping of different types of small particles and accordingly contains a wide mix of unrimed hydrometeor types also in the P17 classification. Lastly, class 16 consists mostly of columnar crystals, of which there are very few in the other classes.

### 5.2.5 Effectiveness of distribution-based clustering

The MASC instrument captures images of falling snowflakes using multiple cameras simultaneously. We did not make use of this capability while training the GAN, and instead operated with single images, because there is often only one sharp image of a given snowflake and requiring multiple high-quality images of each snowflake would have severely limited the size of our training dataset. However, we can use this capability to evaluate the classification scheme because, ideally, the same snowflake viewed from different angles should result in the same classification.

For each snowflake with multiple angles available, we computed the Bhattacharyya distance $d_{\mathrm{B}}$ between the latent-space distributions (obtained as described in Sect. 4.2) of a pair of images of the same snowflake, and the mean of the Bhattacharyya distances to all snowflakes in the dataset. For comparison, we computed the same values for the SED (Eq. 16) between the latent codes obtained without augmentation. Between two images from different angles, median $d_{\mathrm{B}}$ was $1.00$, while median $d_{\mathrm{B}}$ between two different snowflakes was $18.2$ (a ratio of $0.055$). The corresponding median values for the SED are $14.3$ for matched pairs and $20.2$ for all snowflakes (ratio $0.71$).

Another way to compare the two distance metrics is through the distance rank of a pair of images of the same snowflake, which we define here as the percentage of all snowflakes whose distance from the given snowflake is longer than the distance between the pair. The median distance rank is $98.5\%$ for $d_{\mathrm{B}}$ and $70.3\%$ for SED.





Clearly, using the data augmentation and the distribution distance metric in the latent space brings images of the same snowflake much closer to each other. Consequently, we can expect that this approach significantly increases the reliability of unsupervised classification using the latent variables.

## 6   Summary

MASC instruments have been deployed in diverse locations around the world over the last decade, resulting in datasets com-
prising millions of high-resolution images of falling snowflakes. Automated analysis is needed to explore such large quantities of data, and advanced image-processing techniques are beneficial because the image structure contains a signature of the microphysical processes that led to the formation of each snowflake. In this work, we have described an unsupervised approach to this problem using a combination of GANs and $K$-medoids classification. The trained GAN is used to map each image into a vector of latent variables that capture the essential properties of the image. The GAN also learns to generate artificial images
of snowflakes, which we can use to verify that the latent variables map to the properties of snowflakes in a meaningful way. The $K$-medoids algorithm is then used to classify the images based on the latent variables, and the number of classes can be reduced to the desired granularity using hierarchical clustering.

The latent variables code also for information about the images that we do not want to use for classification, such as the orientation. We mitigate this problem by associating each image with a distribution of latent-space points using data augmen-
tation, and defining the distance between images using a distribution-distance metric, the Bhattacharyya distance. Using the multiple cameras provided by the MASC, we verified that this results in improved distance estimates between images, and consequently more accurate classification.

A qualitative assessment of the classification results confirms that each class designated by the classification scheme contains snowflakes with microphysical and structural properties similar to other members of the class. Aggregate snowflakes, which
make up the majority of the dataset, are divided up into several classes, and the differences between these classes reflect the size and the degree of riming of the aggregates. Columnar crystals and small graupel are also quite well separated from other types of ice particles. The hierarchical clustering results in three main branches that differ from each other mostly in the size of the snowflakes.

Each of the applied methods is unsupervised, and consequently can be applied without providing labeled training data.
Hence, the main advantage of the methodology over supervised classification is that the process can be repeated for new datasets with modest manual effort, albeit at a fairly high computational cost. The unsupervised approach also reduces the role of human experts on the classification. This can have both positive and negative effects, as the effect of human biases is reduced, but simultaneously the potential benefits of expert domain knowledge, i.e. understanding of ice microphysics, are neglected. In practice, we find that our classification approach can help distinguish snowflakes by their microphysical properties, but
subsequent analysis is needed to interpret the contents of each class in a microphysical context. Thus, the responsibilities of the domain expert are shifted from creating the training datasets to the less onerous task of interpreting the classification results. The number of classes must also be selected manually, and the class boundaries are somewhat arbitrary as the latent data are



not strongly clustered. Therefore, in the future it may be be interesting to investigate more continuous classification schemes rather than the discrete classification we have described here.

While our approach to unsupervised classification is based on well-documented machine-learning techniques and algorithms, we believe that the combination of methods used here — particularly the use of data augmentation to improve the accuracy of classification using GAN-derived latent variables — has not been employed in previous work. We expect that the same methodology can be adapted to the unsupervised classification of many other datasets in different domains.

*Code and data availability.*   The code and the data supporting this project are available at https://github.com/jleinonen/snow-gan-classification.
This repository includes the training datasets, pre-trained models, derived latent variables and code sufficient to replicate the results.

*Author contributions.*   JL and AB formulated the project and developed the methodology used in this study. JL wrote the software code needed to implement and evaluate the method. JL wrote the article with contributions from AB.

*Competing interests.*   The authors declare that they have no conflicts of interest.

*Acknowledgements.*   The authors acknowledge the financial support from the Swiss National Science Foundation (grant #200020_175700).
The computational resources used in this work were supported by a grant from the Swiss National Supercomputing Centre (CSCS) under project ID s942. We thank Yves-Alain Roulet and Jacques Grandjean of MeteoSwiss for the MASC data from Davos, and Christophe Praz for assistance with processing the data.




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
