# Peer review of "Unsupervised classification of snowflake images using a generative adversarial network and *K*-medoids classification"

_Atmospheric Measurement Techniques, 2019_

## Referee Comment (RC1) · Anonymous Referee #1 · 14 Jan 2020

The authors present a method for unsupervised classification of snow particle images obtained by the Multi-Angle Snowflake Camera (MASC) and demonstrate the ability of the resulting classifications to distinguish features of the snow particles. The topic is relevant - there is a need to be able to analyze snow particle imagery in a mostly auto-mated fashion and to relate the imagery to the microphysical processes that produced a given snow particle. The work is well-structured and presented clearly aside from a some particular details noted below that are related to the mathematical details of the algorithm. Aside from these concerns, the method and assumptions are clear and valid and the results sufficient to support the principal conclusion: that the unsuper-vised method can distinguish snowflakes based on size, shape and texture, and that

the performance of the unsupervised method is reasonably consistent with that of a more labor-intensive supervised method. This unsupervised method has application beyond the MASC to include other imaging disdrometers and particle probes.

My principal concerns are related to some inconsistencies and lack of clarity in the description of the method. My specific comments for this are as follows:

Line 66: Can you clarify the meaning of "deep" CNNs?

Line 78: Can you clarify the meaning of "latent variables"?

Line 108: How does a simple constraint on the diameter of an imaged snowflake ensure that the entire snowflake is within the image frame? Couldn't a snowflake that intersects the edge of the frame have a diameter in this range?

Line 115: What is the specific purpose for downsampling? Is it simply to make the classification processing more computationally tractable?

Line 153-155: It's not clear how "neighborhoods" are defined in the context of a set of snowfall image inputs. Can you elaborate?

Lines 183-184: What is "z"? (see also the comment regarding line 264 below)

Lines 221-222: Is it actually true that the distance between each point and its nearest centerpoint is minimized? I don't believe that is what is imposed by equation 14. But what does lowercase "n" represent in equations 14 and 15?

Lines 260-262: It would help here to have some additional context describing the purpose of a styling block. What is achieved by upscaling the image and processing it through the AdaIN, activation and convolution layers? What is gained by upscaling the image?

Line 264: Earlier (line 211), z is described as the latent distribution. Here it is described as noise, and this seems inconsistent. Can you clarify? What is the difference between "style" and "latent variable"?

Line 340: Does it not appear that there is a threshold near K=3, 4 or 5? There seems to be a substantial change in the slope of the loss function near these K values. Why would this not be seen as an indication of the actual number of medoids?

Finally, I have two technical comments:

Figure 1: I believe the caption is wrong. Panel (a) appears to be the discriminator, and panel (b) the generator.

Line 232: Should this be "understood as a variant"?
* * *

---

## Referee Comment (RC2) · Anonymous Referee #2 · 30 Mar 2020

Title: Unsupervised classification of snowflake images using a generative adversarial network and K-medoids classification

Authors: Leinonen and Berne

A new snowflake classification method using unsupervised machine learning methods is described in this study. It uses approximately two million snowflake images obtained with the Multi-angle snowflake camera (MASC) at various observational sites. This type of method developed will probably be used more and more by the scientific community producing enormous amount of precipitation particle photos using the MASC or other future technology. The paper is well and clearly written. The goal is stated clearly

and the methodology is described in detail. It is, however, difficult for me to evaluate the choice of algorithms used. I only have a few minor comments divided into main comments and specific comments. They are listed below.

Main comments

1. Section 4: The unsupervised methodology developed to analyse the snowflake photos uses the K-medoids method instead of the K-means. These two techniques are described in section 3.4 but the author decided to use only one of them. I was wondering why the K-means is even described in the manuscript. It may be more straightforward to have only a paragraph describing the advantages using K-medoids methods with respect to the K-means and only describe the one used in the developed unsupervised classification.

2. Section 5: The authors describe the methodology used to classify the snowflakes using many K categories. In section 5.2.2, it shows that using 16 classes is more advantageous than using only 6. The authors demonstrate the feasibility and the quality of this unsupervised classification method. In section 5.2.4, it compares the unsupervised classification presented in section 5.2.2. It is difficult to see the link between sections 5.2.2 and 5.2.4 with section 5.2.3. Section 5.2.3 suggests that if we want to analysis microphysical properties of the snowflakes, one may want to have an expert doing it manually. Then, a list of different categories is given. This gives the impression that the method developed is not unsupervised while the goal of that section is probably simply to give an explanation of the limitation associated with the new unsupervised method developed. One suggestion would be to include section 5.2.3 into a discussion provided in the following section.

Specific comments

1. Line 33-35: It is mentioned that "snowflake imagining instruments have been actively developed in the recent years". I am just curious to know if other instruments similar than the MASC exists.

2. Verb tense: The authors should be consistent with the verb tense used in the manuscript. For example, line 122-124 should be past tense. Please verify throughout the manuscript to make sure that it is consistent.

3. Paragraph starting line 270: verb tense please double check.

4. Line 279-281: "However, we found that the latent... classification" should be clarified.

5. Line 283-297: please double check verb tense.

6. Paragraph starting line 323: please double check verb tense.

7. Figures 6, 7 and 9: Does the color code represent the same variable as the number in each square? I think that it is useful to have both the number on each square and the colorbar should be clarified in the figure caption.

8. Section 5.2.3 should probably be included in a following section comparing the unsupervised with the supervised methods.

---

## Author Comment (AC1) · 15 Apr 2020

We thank Referee 1 for their comments that helped clarify a number of key points in the paper. Please find our response below, with original comments quoted in *italics* and our answers and explanations of the changes made to the manuscript in normal font.

*Line 66: Can you clarify the meaning of "deep" CNNs?*

"Deep" in neural network terminology refers to a network with many successive layers. We have now defined this above in the paragraph beginning with "The development of convolutional neural networks..."

[Figure]

*Line 78: Can you clarify the meaning of "latent variables"?*

In GANs that recover the noise variables using the discriminator, the noise becomes semantically a set of latent variables encoding the properties of the input dataset. We now explain this in some more detail.

*Line 108: How does a simple constraint on the diameter of an imaged snowflake ensure that the entire snowflake is within the image frame? Couldn't a snowflake that intersects the edge of the frame have a diameter in this range?*

The images that the MASC takes are actually very big compared to the size of the snowflakes. The P17 processing crops snowflakes from these images such that the snowflakes are completely contained in the frame (snowflakes intersecting the edge of the original image are rejected). We neglected to mention this in the submitted version, it is now explained in this paragraph.

*Line 115: What is the specific purpose for downsampling? Is it simply to make the classification processing more computationally tractable?*

Yes, the main reason is to reduce the computing burden. We believe that this can be done without losing too much information because the MASC images are usually at least slightly blurry and thus the true resolution of the images is not quite as good as the pixel resolution. We have added this explanation here.

*Line 153-155: It's not clear how "neighborhoods" are defined in the context of a set of snowfall image inputs. Can you elaborate?*

This explanation is perhaps confusing so we have reworded it, avoiding the use of the ambiguous term neighborhoods: Pooling layers reduce the spatial dimensions of their input by dividing it into $M \times N$ (typically $2 \times 2$) rectangular region arranged in a grid, then applying a pooling operation such that each rectangle is mapped to a single value in the output image. Usually, either the average or the maximum of the rectangle is used as the pooled value. Pooling operations can sometimes be replaced by strided

convolutions, which skip some points (e.g. every other point) of the input to reduce the spatial dimensionality of the output.

To be clear, the input to a pooling layer is usually not a snowflake image as such, but instead an intermediate stage of processing in the CNN.

*Lines 183-184: What is "z"? (see also the comment regarding line 264 below)*

The variable $z$, the input to the generator (i.e. the noise / latent variable), is defined at the end of the previous paragraph. We have added a note there that $z$ denotes the noise.

*Lines 221-222: Is it actually true that the distance between each point and its nearest centerpoint is minimized? I don't believe that is what is imposed by equation 14. But what does lowercase "n" represent in equations 14 and 15?*

The meaning of this sentence was a bit ambiguous and it has been reworded.

The lowercase n represents "nearest". $c_{n,i}$, as defined in Eq. 15, is the centerpoint closest to the data point $y_i$. This has been reworded for clarity.

*Lines 260-262: It would help here to have some additional context describing the purpose of a styling block. What is achieved by upscaling the image and processing it through the AdaIN, activation and convolution layers? What is gained by upscaling the image?*

The original feature maps that the generator starts with are $4 \times 4$ pixel size and encode the image semantically on a feature level, as learned by the network. To transform the feature maps into a $128 \times 128$ image, we need to:

1. Increase the image size; this is achieved by upscaling the image by a factor of two at a time.

2. Process the deep feature-level representation into an image through a series of

intermediate-level representations; the activation–convolution–AdaIN operations are responsible for these transformations.

We have rewritten the paragraph in a way that hopefully makes this somewhat clearer. A further explanation of the convolution and activation layers can be found in Sect. 3.1.

*Line 264: Earlier (line 211), z is described as the latent distribution. Here it is described as noise, and this seems inconsistent. Can you clarify? What is the difference between "style" and "latent variable"?*

Admittedly, the use of $\mathbf{z}$ in the manuscript was somewhat inconsistent. We have tried to clarify this in the revised version as follows: The generator input, as a whole, is called $\mathbf{z}$; this consists of a latent vector $\mathbf{z}_l$ (which is recovered by the discriminator) and additive noise $\mathbf{z}_a$ (which is not recovered). We have modified this paragraph to specify this and also made changes throughout the article such that this notation is now used consistently.

*Line 340: Does it not appear that there is a threshold near K=3, 4 or 5? There seems to be a substantial change in the slope of the loss function near these K values. Why would this not be seen as an indication of the actual number of medoids?*

There is a change in the slope at nearly every K, but we do not think any of these deviate significantly enough from normal to say unambiguously that they indicate a "correct" number of medoids.

That said, there does seem to be a slightly larger change at $K = 3$ in both the $K$-medoids and in the hierarchical clustering started at $K = 16$ (orange line in Fig. 4). We can get some insight to the nature of that change in the slope by examining Fig. 5. As mentioned in Sect. 5.2.2, the hierarchy in this case consists of three main branches. Therefore, the smaller change in slope at $K > 3$ is likely a result of diminishing returns after the clustering has identified these three main groups. Regardless, the discussion at the end of Sect. 5.2.2 demonstrates that there is a significant benefit in going from

$K = 6$ to $K = 16$, and therefore while $K = 3$ may be optimal in the sense that it provides a large change in slope that indicates the presence of three main groups of snowflakes, it is not a particularly suitable choice for our purposes.

*Finally, I have two technical comments:*

*Figure 1: I believe the caption is wrong. Panel (a) appears to be the discriminator, and panel (b) the generator.*

Thank you, this has been corrected.

*Line 232: Should this be "understood as a variant"?*

Yes, this was also corrected.

---

## Author Comment (AC2) · 15 Apr 2020

We thank Referee 2 for their comments. Please find our responses below in normal font, with the original questions quoted in *italics*.

*1. Section 4: The unsupervised methodology developed to analyse the snowflake photos uses the K-medoids method instead of the K-means. These two techniques are described in section 3.4 but the author decided to use only one of them. I was wondering why the K-means is even described in the manuscript. It may be more straightforward to have only a paragraph describing the advantages using K-medoids methods with respect to the K-means and only describe the one used in the developed*

[Figure]

*unsupervised classification.*

The reviewer is correct in that we do not use $K$-means in this paper. However, we find it is narratively useful to mention $K$-means first, because most readers will be more familiar with $K$-means, and discussing it first allows us to better specify why we choose *not* to use it.

That said, upon reviewing this section we find that there is more technical detail there than is necessary to serve the above-mentioned function. Consequently, we have shortened the discussion of $K$-means and edited the text such that it is clearer that the focus is on $K$-medoids (including changing the section title to "Classification: $K$-medoids").

*2. Section 5: The authors describe the methodology used to classify the snowflakes using many K categories. In section 5.2.2, it shows that using 16 classes is more advantageous than using only 6. The authors demonstrate the feasibility and the quality of this unsupervised classification method. In section 5.2.4, it compares the unsupervised classification presented in section 5.2.2. It is difficult to see the link between sections 5.2.2 and 5.2.4 with section 5.2.3. Section 5.2.3 suggests that if we want to analysis microphysical properties of the snowflakes, one may want to have an expert doing it manually. Then, a list of different categories is given. This gives the impression that the method developed is not unsupervised while the goal of that section is probably simply to give an explanation of the limitation associated with the new unsupervised method developed. One suggestion would be to include section 5.2.3 into a discussion provided in the following section.*

There is an important distinction to be made here between what the expert contributes to the classification process in the different approaches:

- In the supervised approach, such as the P17 scheme discussed in Sect. 5.2.4, the role of the expert is to interpret the training data before it is fed to the machine learning algorithm.

- In our unsupervised approach, the role of the expert is to interpret the nature of the classes after they are created by the machine learning algorithm unlabeled training dataset. In Sect. 5.2.3, we present this interpretation.

In the latter case, the machine learning process is entirely unsupervised and the expert only needs to examine samples from a few (in our case 16) classes. Meanwhile, in the former case the expert needs to manually interpret each item in a training dataset of thousands (in our case, hundreds of thousands) of images. So while the results of the unsupervised classification still need to be interpreted by an expert, this is orders of magnitude less work than in the supervised case. For this reason, we think we can still reasonably call our method unsupervised and we argue that Sect. 5.2.3 should be retained. Meanwhile, 5.2.4 is more of a comparison to earlier work on the same topic than another attempt to interpret the classes using different categories.

We have added text in Sect. 5.2.3 to better explain the rationale for this classification, and in Sect. 5.2.4 to explain that the purpose of that Section is to compare our results to a previously existing classification scheme rather than provide yet another interpretation for our results.

*1. Line 33-35: It is mentioned that "snowflake imagining instruments have been actively developed in the recent years". I am just curious to know if other instruments similar than the MASC exists.*

Yes, there exist some, those we are aware of are the Precipitation Imaging Processor (PIP, also known as SVI or PVI) developed at NASA and the Dual Ice Crystal Imager (D-ICI) from the Luleå University of Technology. We have added mentions and references for these. They have not been commercialized to the extent that the MASC has been, though.

*2. Verb tense: The authors should be consistent with the verb tense used in the manuscript. For example, line 122-124 should be past tense. Please verify throughout the manuscript to make sure that it is consistent.*

Throughout the paper, we have tried to follow the following convention: What we *did* during our experiments is told in past tense, while the description of what the various algorithms *do* and what the datasets contain is told in the present tense (because this is a general description of their content and functionality and not merely what happened once).

Upon proofreading the article, we found some places where, as the reviewer points out, these conventions could have been more consistently followed. We have proofread the entire article and edited to improve this consistency, with particular attention being paid to comments 2, 3, 5 and 6 from this reviewer.

*3. Paragraph starting line 270: verb tense please double check.*

Please see our reply to comment 2 by this reviewer.

*4. Line 279-281: "However, we found that the latent. . . classification" should be clarified.*

We wish the reviewer had been more specific about what is unclear about this paragraph. We have elaborated a bit to make it clearer what the point of this discussion is: That we want to exclude the variation in the latent variables that we do not want to use for classification.

*5. Line 283-297: please double check verb tense.*
*6. Paragraph starting line 323: please double check verb tense.*

Please see our reply to comment 2 by this reviewer.

*7. Figures 6, 7 and 9: Does the color code represent the same variable as the number in each square? I think that it is useful to have both the number on each square and the colorbar should be clarified in the figure caption.*

Yes, this is referred to as a heatmap, see e.g. here for the documentation for the package we used: https://seaborn.pydata.org/generated/seaborn.heatmap.html — the

standard recommended way there seems to be to plot heatmaps with colorbars.

We have now clarified in the figure captions that the figures are heatmaps.

*8. Section 5.2.3 should probably be included in a following section comparing the unsupervised with the supervised methods.*

Please see our response to main comment 2 by this reviewer.

─────────────────

---

## Author Response (AR1)

Dear Editor and Reviewers,

Please find attached our revised version of the manuscript "Unsupervised classification of snowflake images using a generative adversarial network and *K*-medoids classification".

**Detailed response to reviewers:**

We have given point-by-point responses to the reviewer comments in the responses written to the reviewers in the online discussion. Please refer to those responses for a detailed discussion of the points made by the reviewers.

**Changes made to manuscript:**

The reviewer comments have not necessitated major changes to the manuscript. The changes mostly relate to presenting the material more clearly. Some changes required only minor edits, while others required that large parts of sections or paragraphs were rewritten. We have described the changes in detail in our reviewer responses in the online discussion.

**Marked-up changes:**

We have attached below a version of the revised manuscript with the changes highlighted, as produced by *latexdiff*.

[revised manuscript text omitted]

$$
230 \quad L = \frac{1}{N} \sum_{i=1}^{N} d(\underbrace{\mathbf{z}}_{}\mathbf{y}_i, \mathbf{c}_{\text{n},i}) \tag{14}
$$

$$
\mathbf{c}_{\text{n},i} = \underset{\mathbf{c}_j}{\text{argmin}}\ d(\underbrace{\mathbf{z}}_{}\mathbf{y}_i, \mathbf{c}_j) \tag{15}
$$

where $d$ is a distance metric between two points and $\mathbf{c}_{\text{n},i}$ denotes the centerpoint that is nearest to data point $\mathbf{y}_i$. In other words,  the centerpoints are chosen such that the average distance of each point  $\mathbf{y}_i$ to $\mathbf{c}_{\text{n},i}$  is minimized.

235  The standard $K$-means algorithm uses the squared Euclidean distance (SED) metric

$$d_{K\text{-means}}(\underbrace{\mathbf{z}}_{}\mathbf{y}_1, \underbrace{\mathbf{z}}_{}\mathbf{y}_2) = |\underbrace{\mathbf{z}}_{}\mathbf{y}_1 - \underbrace{\mathbf{z}}_{}\
[revised manuscript text omitted]

620